


# Obtaining more benefits from crop residues as soil amendments by application as chemically heterogeneous mixtures

Marijke Struijk[1,2], Andrew P. Whitmore[2], Simon R. Mortimer[3] and Tom Sizmur[1]

[1] Department of Geography and Environmental Science, University of Reading, Reading, UK.
[2] Department of Sustainable Agriculture Sciences, Rothamsted Research, Harpenden, UK.
[3] School of Agriculture, Policy and Development, University of Reading, Reading, UK.

*Correspondence to*: Marijke Struijk (m.struijk@pgr.reading.ac.uk, permanent address: mstruijk@gmx.com)

**Abstract.** Crop residues are valuable soil amendments in terms of the carbon and other nutrients they contain, but incorporation of residues does not always translate into increases in nutrient availability, soil organic matter (SOM), soil
structure, and overall soil fertility. Studies have demonstrated accelerated decomposition rates of chemically heterogeneous litter mixtures, compared to the decomposition of individual litters, in forest and grassland systems. Mixing high C:N ratio with low C:N ratio amendments may result in greater carbon use efficiency and non-additive benefits in soil properties (i.e. mixture ≠ sum of the parts).

We hypothesised that non-additive benefits would accrue from mixtures of low-quality (straw or woodchips) and high-
quality (vegetable-waste compost) residues applied before lettuce planting in a full-factorial field experiment. Properties indicative of soil structure and nutrient cycling were used to assess benefits from residue mixtures, including soil respiration, aggregate stability, bulk density, SOM, available and potentially mineralisable N, available P, K and Mg, and crop yield.

Soil organic matter and mineral nitrogen levels were significantly and non-additively greater in the straw-compost mixture compared to individual residues, which mitigated the N immobilisation occurring with straw-only applications. Addition of
compost significantly increased soil available N, K and Mg levels. Together, these observations suggest that greater nutrient availability improved the ability of decomposer organisms to degrade straw in the straw-compost mixture.

We demonstrate that mixtures of crop residues can influence soil properties non-additively. Thus, greater benefits may be achieved by removing, mixing, and re-applying crop residues, than by simply returning them to the soils *in situ*.

## 1. Introduction

Intensive agricultural systems, with a monoculture of crops and relying on external inputs of fertilisers and pesticides/herbicides, are criticised for their negative environmental impacts. These include the degradation of soil – particularly degradation of soil organic matter (SOM), biodiversity loss, and over-application of N and P (Malézieux *et al*., 2009; Tilman *et al*., 2002). Implementation of multispecies cropping systems (e.g. Malézieux *et al*., 2009) and increasing functional diversity via trait-based approaches (Garnier and Navas, 2012) are some methods that have been proposed to
increase biodiversity and functional complementarity of the variety of species present in arable cropping systems. These approaches can lead to more sustainable nutrient cycling, reduced soil erosion, stabilised crop production, and improvements to a system's innate capacity to resist pests, diseases and other environmental disturbances (Gurr *et al*., 2003). However, some farming systems prevent the cultivation of more than one crop in a field at any one time, and so applying mixtures of crop residues may provide an alternative route to obtaining the benefits of multispecies cropping within monocultural arable
cropping systems.

Crop residues comprise the majority of plant materials harvested worldwide (Medina *et al*., 2015; Smil, 1999) and are readily available on arable farms. Containing carbon and other nutrients, they present a valuable resource as soil amendments with the potential to increase SOM and nutrient levels, which feed the soil food web (Kumar and Goh, 1999) and may increase soil aggregation and improve soil structure (Cosentino *et al*., 2006; Martin *et al*., 1955). Unfortunately,


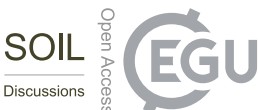

while these changes in soil properties are likely to lead to increased crop yield, decomposition of residue soil amendments does not always translate into such benefits and is instead followed by loss from the system, with lower soil N retention and C levels than expected (Catt *et al.*, 1998; Powlson *et al.*, 2011; Thomsen and Christensen, 2006).

Rather than applying a single crop residue, mixtures of crop residues could form a better soil amendment. Complementarity in mixtures of different residues has been previously shown in research on the decomposition rates of mixtures of moss and

leaf litters in forest ecosystems and grass clippings in grassland ecosystems (Gartner and Cardon, 2004; Hättenschwiler *et al.*, 2005). Synergistic non-additive mixing effects are frequently observed, i.e. decomposition of the mixture is greater than would be predicted from the rate of decomposition of individual litter types (mixture > sum of the parts), especially when the litters are chemically heterogeneous (Pérez Harguindeguy *et al.*, 2008; Wardle *et al.*, 1997).

Suggested mechanisms for non-additive decomposition rates of mixtures include physical, chemical and biological processes

(Gartner and Cardon, 2004). Frequently cited is the mechanism that N-rich residues are thought to accelerate the decomposition of N-poor residues (Seastedt, 1984) by inter-specific transfer of nutrients in the residue mixture (Berglund *et al.*, 2013; Briones and Ineson, 1996). Additionally, more heterogeneous and improved micro-environmental conditions increase habitat and resource options for decomposer organisms (Hättenschwiler *et al.*, 2005), also known as the improved micro-environmental condition theory (Makkonen *et al.*, 2013).

However, whether synergistic decomposition rates in mixtures are related to benefits in terms of soil nutrient and carbon management is unclear because studies on the C and N dynamics in decomposing residue mixtures are limited (Redin *et al.*, 2014). It has been shown that increased plant species richness can promote soil C and N stocks via higher plant productivity (Cong *et al.*, 2014) and to increased diversity and functionality of soil microbes (Lange *et al.*, 2015) as well as the whole soil food web (Eisenhauer *et al.*, 2013). Quemada and Cabrera (1995) found non-additivity in the C and N dynamics when

mixtures of leaves and stems were decomposed compared to individual residues, with the C:N ratio of the residues playing an important role in N mineralisation. Nilsson *et al.* (2008) report synergistic effects on soil available N as well as on plant productivity when mixing *Populus tremula* litter (C:N = 40, known to decompose quickly) with *Empetrum hermaphroditum* (C:N = 77, known to decompose slowly). These experiments suggest that non-additivity in decomposition rates and changes to other soil properties could go hand-in-hand.

Increasingly more evidence is emerging that SOM accumulation is primarily derived from the production of microbial residues (Ludwig *et al.*, 2015; Simpson *et al.*, 2007), and this microbially-derived SOM seems to be produced at the early stages of plant-residue decomposition (Cotrufo *et al.*, 2015). Microbial carbon use efficiency (CUE) describes a functional trait of microbes that refers to the fraction of carbon assimilated from organic matter additions to the soil system compared to C losses to the atmosphere via microbial respiration (Allison *et al.*, 2010). Different microbial species exhibit an inherent

CUE window, so that they can operate at different CUE levels to fulfil their maintenance and growth C requirements depending on environmental factors (Schimel *et al.*, 2007). Organic substrates can feed into different microbial metabolic pathways (e.g. anabolism vs. catabolism) or microbial communities that exhibit different overall inherent CUE levels (e.g. fungi vs. bacteria, or copiotrophs vs. oligotrophs) (Jones *et al.*, 2018). Therefore, an increase in the amount of SOM from microbial activity is not linearly related to $CO_2$ production, or to the quantity of C applied to the soil, but depends also on the

CUE of the decomposer community.

Fertilisation practices typical of intensively managed arable soils stimulate copiotrophic microorganisms (Fierer *et al.*, 2012) with boom-bust population dynamics. These microbial communities tend to exhibit a lower inherent CUE window than slower growing oligotrophic communities (Ho *et al.*, 2017; Roller and Schmidt, 2015). In intensively managed arable soils, the decomposition of soil-applied crop residues can lead to a large portion of residue-derived C being respired as $CO_2$ rather

than turned into SOM (Bailey *et al.*, 2002; Six *et al.*, 2006). Decomposition of high-C:N residues requires microbes with a relatively high CUE, but due to N-limitation they operate towards the lower end of their CUE window (Kallenbach *et al.*, 2019). Low-C:N residues, providing relatively more N, may increase the CUE of individual microbes, but can also shift the





composition of the soil microbial community to one that exhibits an inherently lower CUE (Kallenbach *et al.*, 2019). As suggested by Kallenbach *et al.* (2019), a mixture of crop residues of different C:N ratios could therefore achieve a more

diverse microbial community comprising organisms fulfilling niches of both high and low inherent CUE windows, and may enable all species to operate at their maximum CUE. Other authors have also suggested the possibility of manipulating the functionality of the soil microbial community with soil amendments, such as Li *et al.* (2019) who report that microbes in a eutrophic system are stimulated by organic carbon amendments and oligotrophic microbes are stimulated by chemical fertilisers. Studies have also demonstrated that changes in tree litter diversity affect both fungal and bacterial diversity

(Otsing *et al.*, 2018; Santonja *et al.*, 2018).

Low-quality plant materials with high C:N ratios constitute the majority of crop residues produced by arable farming practices worldwide, typically involving cultivation of corn, wheat and rice (Medina *et al.*, 2015). The potential of crop residue soil amendments to deliver benefits to crops would be better exploited if the decomposition processes were manipulated for C to persist in the soil biomass, necromass or other forms of (semi-)stabilised SOM, such as in soil

aggregates. Generally soil amendments consisting of one large amount of a single crop residue do not always deliver benefits. We suggest that the non-additive decomposition rates observed in forest litter mixtures reinforced by recent insights into the link between CUE and the difference in C:N ratio of soil organic co-amendments, can inform strategies to obtain more benefits from crop residues as soil amendments. Mixing these crop residues to create chemically diverse crop-residue mixtures with a CUE-optimised C:N ratio to generate a greater diversity of functionally complementary microbial niches and

to enable each member of the microbial community to function at a maximised CUE, could be a relatively simple method to obtain more benefits from this precious, but ubiquitous, resource. If this approach can attain higher CUE levels for high-C:N residues, a considerable increase in net SOM could be realised in arable cropping systems, along with other beneficial changes in soil properties (e.g. nutrient retention) leading to greater soil fertility, and meanwhile increasing biodiversity in otherwise monocultural arable cropping systems.

The aim of this study was to investigate the potential of chemically heterogeneous mixtures of crop residue amendments to improve soil properties for crop production. A field experiment was set up on an intensive organic arable cropping farm. Amendments of mixtures and individual crop residues were applied: vegetable waste compost was used as low-C:N (high-quality) residue, and wheat straw and woodchips were used as high-C:N (low-quality) residues. Properties indicative of soil structure and nutrient cycling were used to assess benefits from residue mixtures compared to individual residues, including

lettuce crop yield, soil respiration, soil aggregate stability and bulk density, SOM, available and potentially mineralisable N, and available P, K and Mg. We predicted higher decomposition rates when mixtures of crop residues were applied compared to individual residue amendments, leading to non-additive effects in soil properties that could be beneficial for crop production. In particular, we hypothesised faster decomposition of residue mixtures to result in a higher soil respiration rate in the short term, as well as the release of greater levels of soil available nutrients (N, P, K, Mg) and SOM compared to

individual residues (hypothesis 1). An increase in SOM will likely change soil physical properties, which we expected to observe as an increase in soil aggregate stability and a decrease in soil bulk density (hypothesis 2). These changes in soil physicochemical properties were subsequently expected to lead to a higher crop yield (hypothesis 3).

## 2. Methodology

### 2.1. Study site and experimental design

A field experiment was set up in an intensively managed horticultural area of lowland fen on an organic farm near Ely in Cambridgeshire, United Kingdom (52° 21' N; 0° 17' E). During the experiment, between 11 June 2018 and 26 July 2018, the field site was used for growing gem lettuce crops (*Lactuca sativa* L. var. longifolia, commercial variety 'Xamena'), following a year of celery crop in 2017, conversion to organic in 2016 (grass ley), winter wheat in 2015, and beetroot in





2014. The typical crop rotation followed by the farm is celery, followed by beetroot, celery or onion, followed by lettuce,
followed by a break crop of perennial ryegrass and white clover or a cereal. The experimental plots were located on clay
loam, on a roddon, a dried raised bed formed by the deposition of silt and clay from a watercourse which pushed peat to the
sides. The mineral part of the soils typically do not perform as well as the surrounding organic soils because they require
more fertiliser, so we expected they would respond more quickly to residue amendments.

Four replicates of six treatments, within a full-factorial randomised complete block design of the factors *compost* and *residue*
(Table 1) were applied to 2 m × 6 m experimental plots within a 6 m × 48 m field site consisting of 3 × 8 = 24 plots situated
between the tire tracks of farm machinery. All samples were taken from the inner 2 m × 2 m of each plot to incorporate a 4-
metre long buffer zone between plots along the same strip.

The residue amendment treatments were prepared on 17 May 2018. Application rates of the different amendments were 20 t
ha⁻¹ fresh compost (equivalent to 7 t ha⁻¹ dry matter), 13.3 t ha⁻¹ woodchips (equivalent to 8.7 t ha⁻¹ dry matter) and 10±0.8 t
ha⁻¹ straw (equivalent to 9.2±0.8 t ha⁻¹ dry matter). These are within the range of application rates that are common in
intensive arable cropping systems in Europe (Recous *et al.*, 1995; S. Gardner, 2018, pers. comm.), and were chosen to obtain
similar amounts of dry matter for each residue. These rates were consistently applied in both individual amendment
treatments and mixtures, so residue-compost treatments contained roughly twice as much dry matter compared to individual
amendments. Applications were spread out evenly over the plots by hand on 12 June 2018 (Figure 1c), followed by power-
harrowing to incorporate the residues in the soil profile. Gem lettuce plugs were sown the following day.

**2.2. Soil and residue characterisation**

Baseline soil samples were collected on 11 June 2018 (before organic amendments were applied). For each plot, soil samples
were collected as the combination of five 30 mm diameter soil cores taken to 20 cm depth. These 24 composite samples
were air-dried, disaggregated with the aid of a mortar and pestle, sieved to 2 mm and analysed for soil moisture (at 105 ºC
overnight), SOM by loss on ignition (LOI) (at 500 ºC overnight), pH (after 2 hrs shaking 2.5 ±0.005 g soil with 25 ml
Ultrapure water [> 18.2 Ω/cm]), and soil texture by laser granulometry (Malvern Mastersizer 3). A portion of each soil
sample was ball milled and analysed for total C and N (Flash 2000, Thermo Fisher Scientific, Cambridge, U.K., calibrated
with aspartic acid, 104% N and 100% C recovery rates of in-house reference soil material traceable to GBW 07412). There
were no significant treatment differences for any of these baseline soil variables, tested with a one-way analysis of variance
(ANOVA) of treatments or a two-way ANOVA of the factors *residue* and *compost* (Table 2).

All amendments were provided by the farm and sourced and prepared on-site. The compost amendment was composed of
the following vegetable residues from the farm: spinach, celery, several lettuce varieties, carrots, leeks, spring onions, onions
and shallots, cabbage, bell peppers, beetroots, and mushrooms (Figure 1a-b). Due to the high water content of these residues,
the farm co-composts with straw to provide sufficient dry matter content in the compost mixture. The straw amendment used
in the treatments containing straw was winter wheat straw available on-site, and the woodchip amendment was from poplar
trees commonly grown as a wind break in the local area. Dried and milled residues were analysed for total C and N (Flash
2000 as aforementioned, 109% recovery rate of both C and N of in-house reference rapeseed material, traceable to certified
reference material GBW 07412). The total concentrations of P, K and Mg were determined by ICP-OES (inductively
coupled plasma optical emission spectroscopy, Perkin Elmer Optima 7300 Dual View, 99%, 94% and 102% recovery rate of
160 P, K and Mg, respectively, of in-house hay reference material traceable to certified reference NCSDC 73349) analysis of
0.5 g residues samples digested in 8 ml of nitric acid (trace metal grade) using MARS 6 microwave digestion system (Table
3).

The amounts of C, N and other nutrients applied in each treatment were calculated based on the chemical characterisation of
the residues and their application rates (Table 4).



### 2.3. Assessment of yield

Lettuce crops were planted on 14 June 2018 and harvested from the inner 2 m × 2 m of each plot on 20 and 21 July 2018, i.e. 38 days after residue application and 36 days after planting. Each lettuce head was harvested whole and weighed to calculate the total biomass produced per treatment. Meanwhile lettuce crops were qualitatively assessed, which included screening for chlorosis, caterpillar damage, tip burn, and rotting. In some cases dried out mushrooms were found on the outer leaves, which was also noted.

### 2.4. Assessment of soil biogeochemical properties

All soil samples were taken from the inner 2 m × 2 m of each plot on 26 July 2018, i.e. 44 days after residue application. From each plot a 10 cm deep bulk density core of 9.8 cm diameter was collected. A series of six 30 mm diameter soil cores to 20 cm depth were collected, combined and homogenised in a zip-lock bag, and used for a suite of analyses. A sub-sample of the fresh soil was sieved to 2 mm for analysis of available N (i.e. sum of $NO_3^-$ and $NH_4^+$) by 1 M KCl extraction before and after a 4-week incubation at 70% of the water-holding capacity (WHC). Extracts were filtered through a Whatman no. 2 filter and analysed colorimetrically for $NO_3^-$ and $NH_4^+$ on a Skalar San$^{++}$ continuous flow analyser. *Available N was taken as the sum of the* $NO_3^-$ *and* $NH_4^+$ *measured in the first extract. Potentially* mineralisable N was calculated as the difference in $NO_3^-$ and $NH_4^+$ measured before and after the 4-week incubation period. A sub-sample of the fresh soil was sent to NRM laboratories (Bracknell, UK), where it was air-dried and sieved to 2 mm for measurement of available P by extraction with 0.5 M $NaHCO_3$, available K and Mg by extraction with 1 M $NH_4NO_3$, soil particle size distribution by laser granulometry, SOM based on LOI at 430 ºC, and the Solvita $CO_2$ burst test measuring the concentration of $CO_2$ produced by soils moistened to 50% of their WHC.

Earthworm and mesofauna sampling was performed, but only a few juvenile earthworms were found, which made identification difficult. The endogeic species *A. chlorotica* was identified in at least three of the 24 plots. The abundance of mesofauna (Collembola and mites) extracted from the soils using Tüllgen funnels was null. Some Collembola were observed while harvesting the lettuce crop, so their absence from the samples is probably due to the removal of plants that provided some shelter from the hot and dry weather conditions.

Wet aggregate stability was assessed as per Nimmo and Perkins (2002) using soil samples that were collected into tubs (to prevent soil compression) from the top 10 cm of each plot, and subsequently air-dried. A 4 g subsample from each plot was slowly pre-wetted on moistened filter paper. The wet sieving procedure involved a wet-sieving apparatus composed of vertically moving 250 µm sieves to hold the soil samples sitting inside a can. The cans were filled up with water such that the soil was submerged, causing the unstable soil aggregates to break apart and pass through the sieve into the can. First, the soils were wet-sieved for 3 minutes in deionised water to collect unstable soil particles and subsequently in a solution of 2 g/L $(NaPO_3)_6$ to disperse the water-stable aggregates. The stable fraction of soil (i.e. wet aggregate stability) was then calculated as the weight of soil caught by the dispersing solution divided by the sum of the weights of soil caught by both water and dispersant. Any particles larger than 250 µm did not pass the sieve and were not included in the calculation.

### 2.5. Data analyses

We observed a gradient in the soil %C and a similar gradient in the %N content of the baseline soil samples that was not well captured by our original blocking design, so the data were retrospectively blocked accordingly (Supplement S1). This was necessary because the calculation of non-additive effects, described below, relies on paired samples within blocks rather than treatment averages across blocks.

Statistical analyses were performed in R 3.5.1 (R Foundation for Statistical Computing) using RStudio 1.1.456 (RStudio, Inc.). To determine effects of treatments and/or factors on individual soil parameters, a two-way ANOVA, including interactions, with the factors *compost* (compost or no compost) and *residue* (straw, woodchips or no residue) was performed.





If a factor had a significant effect (p < 0.05), a post-hoc Tukey HSD test was run to determine which treatments were significantly different from each other. Taking into account that four replicates per treatment is a limited number of data points, assumptions of the ANOVA test were assessed both visually and via the relevant statistical tests: homoscedasticity was evaluated with a Q-Q plot of the ANOVA residuals plotted against the fitted data of the ANOVA, as well as a Levene

test of the data set. Normal distribution of the residuals was evaluated with a residuals-versus-fitted plot and a Shapiro-Wilk test of the residuals of the ANOVA. Pearson correlations were performed to investigate relationships between different variables.

Properties indicative of soil structure and nutrient cycling were used to assess non-additive effects from residue mixtures compared to individual residues, including lettuce crop yield, soil respiration, soil aggregate stability and bulk density, SOM,

available and potentially mineralisable N, and available P, K and Mg. The % effect of each measurement of the treatment effects was first determined by adjusting to the measured effect of the control treatment:

$$\% \; effect = \frac{treatment - control}{control} \; 100\% \tag{1}$$

Next, the % non-additive effects of the residue mixtures were calculated as the difference between the % effect of the mixture and the % effect of the sum of the parts:

$$\% \; non-additive \; effect_{mixture} = \% \; effect_{mixture} - (\% \; effect_{compost} + \% \; effect_{residue}) \tag{2}$$

where *residue* refers to straw or woodchips. A one-sided T-test of the % non-additive effects was performed with an alternate hypothesis (H₁) of μ > 0 for yield, available N, potentially mineralisable N, available P, K, Mg, soil respiration, SOM, aggregate stability, and an alternate hypothesis of μ < 0 for bulk density and pH. Normality was tested with a Shapiro-Wilk test.

## 3. Results


### 3.1 Non-additive effects

Non-additive effects measured 44 days after application of the treatments were mostly synergistic (i.e. mixture > sum of the parts), although the majority of effects were not statistically significant (Figure 2). The magnitude and direction of deviation from additivity were usually similar for both the woodchip-compost and straw-compost mixtures, although non-additive

effects from the woodchip-compost mixture were sometimes less pronounced than those from the straw-compost mixture.

Both compost-residue mixtures resulted in a non-additive increase in lettuce yield, available and potentially mineralisable N, available Mg, SOM, and soil respiration, but not in available K (hypothesis 1), some of which were statistically significant (Table 5). Most notably, we observed greater available N and SOM levels in soils to which a mixture of residues was applied, compared to the available N and SOM levels in treatments receiving only individual residue amendments. The

straw-compost mixture resulted in a significant (T = 4.022, p = 0.014) non-additive increase in SOM of 13.10%, and while the woodchip-compost mixture did not result in statistically significant non-additivity (T = 0.954, p = 0.205), it did result in a positive non-additive increase in SOM of 6.73%.

Likewise, amendment with the straw-compost mixture led to significantly (T = 3.789, p = 0.016) greater available N levels that were 55.06% higher on average than would have been expected from the available N levels in treatments receiving

individual amendments of straw or compost only. The positive non-additive effect on available N observed in soils that received the woodchip-compost mixture was, however, smaller (7.16% increase on average) and not statistically significant (T = 0.235, p = 0.415). A non-significant non-additive increase in available P was only observed after application of the straw-compost mixture, but not after application of the woodchip-compost mixture (hypothesis 1). Contrary to our hypothesis, there was a non-additive increase in pH from the mixtures relative to individual amendments (hypothesis 1),

although this was not significant (Table 5) and per-treatment results (discussed in next section) show that the pH decreased in all treatments relative to the control (F = 2.238; p = 0.095; one-way ANOVA; Supplement S2). We also observed non-





additive effects from both compost-residue mixtures on the soil structure, i.e. a decrease in bulk density and an increase in aggregate stability (hypothesis 2), and a non-additive increase of about 10% was found for crop yield from both crop-residue mixtures (hypothesis 3). Although the effects on soil structure and yield were mostly non-significant, the decrease in bulk

density after amendment with the straw-compost mixture was borderline significant (F = -2.232, p = 0.056) (Table 5).

The following sections contain the per-treatment results of the soil physical and biochemical properties measured in this experiment. It should be noted that application rates of the mixtures were about twice as high as individual amendments to enable calculation of non-additivity, so measurements from residue-mixture treatments cannot be directly compared to individual-residue treatments.

**3.2. Per-treatment results**

Yield assessed by total biomass of gem lettuce produced per plot seemed to be somewhat reduced by the straw-only treatment but was not significantly affected by any of the treatments or factors (Figure 3a; see Supplement S4 for statistical outputs).

Lettuce plants in the straw-only treatments suffered noticeably less damage, particularly from caterpillars, tip burn, and rot
(Table 6). There was a significant interaction between *residue* and *compost* in terms of the qualities of lettuce plants harvested (F = 3.568, p = 0.050; two-way ANOVA), with the biggest difference between straw-only and straw-compost (p = 0.067; post-hoc Tukey HSD). Mushrooms were observed on the outer leaves of some lettuce heads in plots receiving woodchips, or in two cases in plots neighbouring treatments including woodchips, so fungi may have been introduced and/or promoted by woodchips.

Levels of SOM and N (available and potentially mineralisable) were negatively affected by the straw-only treatment, while treatments of woodchip-only and compost-only had little effect on SOM and N levels compared to the control (Figures 3b and 4). Residue mixtures increased SOM and N in most cases, with the exception of the effect of the straw-compost treatment on SOM. Nonetheless, there was a non-additive effect in SOM and N in the straw-compost treatment, as this non-additivity was in fact a negation of the negative effect on SOM and N of straw applied as an individual residue.

Treatment effects on SOM or N levels were not significantly different between treatments (SOM: F = 0.981, p = 0.456; N: F = 1.81, p = 0.163; one-way ANOVA), but the factor *compost* tended to increase soil N (F = 3.88; p = 0.065; two-way ANOVA). Soil respiration in the different treatments was rather similar in all treatments and none of the treatments caused soil respiration to deviate significantly from the control or from each other (F = 1.358, p = 0.286; one-way ANOVA; Supplement S2).

The addition of compost, either as an individual residue or in a mixture, significantly affected soil available K (F = 7.761; p = 0.012) and Mg (F = 4.953; p = 0.039) (Figure 5a). Akin to soil N and SOM, the lowest levels of nutrients were found in soils amended with the straw-only treatment. The increases in nutrient availability were not consistent with the crop residue amendments and ranged from -242% to 57% of the nutrient added as part of the amendments (Supplement S3). If there was an increase in nutrients, the contribution of the amendments was relatively small in most cases and exhibited very large error
margins. The most notable observations from these data is the consistent immobilisation of nutrients brought about by the straw-only treatment, while amendments including woodchips or compost had a tendency to modestly increase soil available nutrients. None of the nutrient increases exceeded 100% of the nutrients added, indicating that residue amendments did not result in net mobilisation of nutrients already present in the soil.

We observed no significant effects on the aggregate stability of the differently amended soils, but the soil bulk density
tended to be lowered by the *residue* factor, i.e. when a low-quality residue was part of the treatment (F = 3.28; p = 0.062; two-way ANOVA) (Figure 5b).



### 3.3. Correlations

A number of noteworthy correlations may help explain the data and are summarised in Table 7. There were some significant correlations between the amount of nutrients applied and the amount of available K and Mg in the soils at the end of the
experiment, which indicates a positive effect of the residue amendments. The amount of C applied via the residue amendments was not correlated with the levels of SOM. Yield was positively and significantly correlated with the sum of available and potentially mineralisable N, available P and Mg, SOM and aggregate stability.

## 4. Discussion

### 4.1. Non-additive effects

The objective of this study was to find out if greater benefits could be obtained from crop-residue soil amendments in an arable soil by applying them as chemically heterogeneous mixtures of low-C:N vegetable waste compost with high-C:N straw or woodchips, compared to individual residue amendments. Relative benefits of the mixtures were assessed by calculating the non-additivity of a range of effects, including yield and a selection of soil properties that are likely to be beneficial for crop production. We found some degree of non-additivity in the direction (synergy or antagonism) we
predicted in most parameters (except available P in the woodchip-compost mixture and available K in both mixtures), and significant non-additive increases in available N and SOM after application of the straw-compost mixture, indicating that even after a short amount of time (44 days) beneficial effects from a mixture of residues can be greater than the sum of its parts.

Examining per-treatment effects can help further explain the non-additivity results. The per-treatment difference in terms of
SOM and available N between the woodchip-compost treatment and the straw-compost treatment was relatively small. Yet, only the straw-compost mixture exhibited significant non-additivity. Comparison of the per-treatment effects on SOM and available N reveals that the significant non-additive effects observed after application of the straw-compost mixture are in fact a negation of the negative (compared to control) effect of the straw-only treatment. As suggested earlier, this indicates that decomposition of single crop residue amendments does not always translate into agronomic benefits, and applying
mixtures of crop residues could be a route to improve those benefits.

### 4.2. Decomposition

Although we suggested that non-additive effects might be related to differences in decomposition rates in the mixtures compared to the individual residues, we have no evidence of this in terms of soil respiration measurements. At the time of sampling, high microbial activity may have increased N immobilisation and therefore decreased soil mineral N availability.
However, respiration rates were equally low in the straw-only (N immobilisation) and the straw-compost treatments (N mineralisation), and both were lower than the control (Supplement S2). Likewise, Redin *et al*. (2014), who studied residue mixtures of stems and leaves of 25 different arable crop species, found mostly additive effects for decomposition rates of mixtures, but unlike the results presented here they found no synergistic effects on N mineralisation. Both here and in the study by Redin *et al*. (2014), decomposition was measured in terms of C mineralisation (measured as $CO_2$ release), which
does not account for the possibility of a higher CUE when chemically diverse residue mixtures are applied, and also does not distinguish between mineralisation of residues or organic matter already present in the soil. Moreover, our soil respiration measurements were taken by the Solvita burst method, on soil samples removed from the field and sieved to 4 mm, which may not have been a good representation of the respiration produced in-situ by a soil mixed with crop residues at various stages of decomposition.

Another reason for the absence of different soil respiration rates may be the relatively short duration of this experiment, covering the short growing period of gem lettuce. As pointed out by Lecerf *et al*. (2011), niche complementarity effects, in





which different groups of decomposing organisms develop a synergistic association in residue breakdown, tend to advance with time, leading to a generally higher number of long-term litter-mixing studies finding non-additive effects. Indeed, Ball *et al.* (2014) only observed a non-additive effect on mass loss in a five-component mixture after 193 days. Therefore an
experiment of longer duration may be able to capture more and greater treatment effects and non-additive effects.

### 4.3. Yield

Although yield, assessed by total biomass of gem lettuce produced per plot, was not significantly affected by any of the treatments or factors, there were some notable differences between treatments. Yield appeared to be somewhat depressed by the straw-only treatment, which is not surprising considering the lower concentration of soil available N, SOM, soil nutrients
and aggregate stability in this treatment, compared to the control. Crops tend to require most nitrogen during the vegetative growth stage and when this is not available, yield will be affected (Chen *et al.*, 2014). The lettuce plants were planted as plugs just after application of the treatments, so when they were introduced to the experimental plots they were already in their vegetative stage. Significant positive correlations of yield with the sum of available and potentially mineralisable N, available P and Mg, SOM, and aggregate stability suggest that these are the main benefits provided by the crop residue
amendments from an agronomic perspective.

Overall lettuce quality was least affected in the straw-only treatments, despite the location of these treatments being towards the low soil-C end of the field site (Supplement S1). Available N levels were positively correlated with overall quality impairment (i.e. % lettuce heads affected by some form of quality impairment) ($p = 0.011$), and in particular with yellow tips ($p = 0.017$) and tip burn ($p = 0.041$), which may indicate the crop was suffering from N deficiency (Table 7). Indeed, the N
levels were relatively low compared to those recommended for lettuce crops (RB209, 2019), and N deficiency leads to reduced plant size, which would lead to decreased biomass production, as well as chlorosis and outside leaves senescing prematurely and dropping off (Brady and Weil, 2002), all of which were observed.

### 4.4. Nutrient dynamics and transfer

The straw-only treatment led to a notable immobilisation of N, which was unlike the other treatments. Although this could
be only a temporary effect (e.g. as in Silgram and Chambers, 2002), it may be unfavourable for lettuce crop productivity and should be taken into account when timing crop residue applications. The notable N immobilisation in the straw-only treatment suggests that straw decomposed differently as an individual residue than in a mixture with compost, which could be explained by the C:N ratio of the treatments. Chen *et al.* (2014) evaluated soil N processing during crop residue decomposition and suggested that residues with a C:N ratio below ~25 result in net mineralisation (increase in soil available
N) and those with a C:N ratio above ~30 result in net immobilisation (decrease in soil available N). Therefore, in the present study the woodchip-only (C:N = 64) and straw-only (C:N = 41) treatments are both expected to result in net N immobilisation. The reason why N immobilisation is only observed in the straw treatment could be due to a lower decomposition rate of the woodchips and therefore lower microbial N-mining requirement at the time of sampling. Straw is likely more decomposable due to a comparatively lower C:N ratio, a higher water-holding capacity (being more friable and
having a greater surface area to hold on to moisture) (Hättenschwiler *et al.*, 2005; Iqbal *et al.*, 2015) and possibly a soil microbial community that is more adapted to decomposing straw because wheat is sometimes grown in these soils.

A slight increase in soil N (available and potentially mineralisable N) observed in the straw-compost treatment and to a lesser extent in the woodchip-compost treatment, compared to the control, could be due to N derived from the compost, the residue, or primed native SOM. Priming of native SOM caused by the amendment seems unlikely in the woodchip-compost
treatment, because SOM levels were higher compared to the control treatment. Even in the straw-compost treatment, the SOM level was very close to that of the control treatment, suggesting mineralisation of native SOM was negligible. Compost was the most significant factor related to higher soil N levels, which can be attributed to its low C:N ratio, allowing for easy





decomposition with minimal immobilisation of native soil mineral N. In the residue mixtures, it is likely that compost provided nutrients for decomposer microbes to be able to decompose the high-C:N residues (i.e. inter-specific nutrient transfer).

Therefore, the non-additive effects on soil N in the straw-compost treatment can probably be attributed to interspecific net transfer of N from high-N to low-N residues resulting in (1) the retention of compost-derived N by straw or woodchips in the mixture, preventing it from being leached, and (2) a higher nutrient availability in treatments including compost, enabling decomposer organisms to break down and release N contained in the amendment mixture more readily. The transfer of N

can occur by a combination of uptake and release by microbes on the high-N residue as they produce enzymes for decomposition, and diffusion along a gradient of high N to low N (Schimel and Hättenschwiler, 2007). The woodchips likely had a higher lignin content than straw. Ligninolytic enzyme production can be inhibited by elevated N concentrations (Carreiro *et al*., 2000; Knorr *et al*., 2005), resulting in a relatively greater inhibition of decomposition of the woodchips.

The transfer of N in litter mixtures appears to go hand in hand with a C transfer. In a microcosm experiment by Berglund *et*

*al*. (2013) on pine and maize litters inoculated with both forest and arable soils, mixing residues mostly increased C loss from the lower quality litter, while C released from the higher quality litter was equivalent to decomposing as an individual litter. Therefore, the non-additively higher SOM in the straw-compost treatment is likely to be the result of enhanced C release from the straw due to the addition of compost. This phenomenon could be explained by a bidirectional transfer of C and N between high- and low-quality residues – e.g. via transport of amino acids by fungal mycelia (Tlalka *et al*., 2007) –

where increased N availability near the low-quality residue enhances its decomposition and subsequent C release, while increased C in the presence of the high-quality residue has little effect on its decomposition (Berglund *et al*., 2013).

### 4.5. Soil physical structure

Increased SOM positively affects aggregate stability because soil microbes feeding on organic substrates enhance soil aggregate formation and stability by biofilm formation and the production of extracellular polymeric substances that increase

cohesion between soil particles (Martens, 2000; Totsche *et al*., 2018). Aggregate stability, in turn, is involved in the protection of mineral-associated SOM (Angst *et al*., 2017). Therefore, with an increase in SOM, an increase in aggregate stability would be expected, and we did indeed observe a positive correlation between these variables ($p = 0.028$). We also observed a positive correlation between aggregate stability and soil available N ($p = 0.005$). This is contrary to the observation that high-quality residues and/or addition of N fertilisers result in higher aggregate turnover (formation and

breakdown) compared to a greater aggregate stability when low-quality residues are applied (Chivenge *et al*., 2011).

Because we observed positive effects on both soil N and SOM from crop residue mixtures, an increased non-additive effect on the soil physical structure from application of the right residue mixtures can therefore be anticipated over time. However, in many arable cropping systems tillage may undermine the emergence of this benefit by destroying soil aggregates and exposing the SOM contained within (Nath and Lal, 2017). Furthermore, bulk density was lowered by the addition of the

low-quality residues (straw and woodchips; $p = 0.062$), especially when combined with compost. This could be partially due to increases in the aggregate stability in most of these treatments, although some residues (with a lower density than soil) may have also been included in the bulk density ring when sampling.

### 4.6. Potential of residue mixing to obtain more benefits from low-quality residues

Our study provides some evidence that chemically heterogeneous crop residue mixtures can provide agronomically

beneficial non-additive effects. We found prevention of N immobilisation to be the most prominent effect in the short term. Positive non-additivity in SOM levels and other soil nutrients may develop over time, but a longer term experiment is necessary to investigate this.

Other authors have also found beneficial effects on soil N levels from mixed residue amendments. For instance, Kaewpradit *et al*. (2009) mixed groundnut residues (high N) and rice straw (low N), which slowed down N loss by mineralisation during

the phase between two different crops, i.e. a beneficial temporary N immobilisation. McDaniel *et al*. (2016) found that non-additive effects of soil C and N dynamics after application of residue mixtures depend on the diversity in cropping history, with non-additive effects primarily observed in monoculture soils rather than diverse crop rotations. The authors attribute this to the low respiration rates from monoculture soils after application of low-quality residues, while soil response to high-quality residues is similar in both monoculture and crop rotation soils (McDaniel *et al*., 2016). These studies indicate that

potential benefits from residue-mixing are dependent on the arable cropping system.

Manipulation of the number of component residues, the mixing ratio, and the quantity applied can be used to optimise timing and amount of nutrient release for a better synchrony with crop demand (Myers *et al*., 1997). For instance Kuo and Sainju (1998) demonstrated that the timing of N mineralisation can be manipulated by the proportion of leguminous cover crop residues in the mixture, while Mao and Zeng (2012) found that both the number of residue components and their mixing

ratio affected non-additivity. Furthermore, the quantity of residues applied can impact on microbial CUE: while microbial CUE is often unaffected at low substrate additions, applications of high amounts of the same material can lead to diminishing CUE levels (Jones *et al*., 2018), e.g. as shown by Roberts *et al*., 2007 with glucose and glucosamine additions to various foraging soil types in a microcosm experiment.

The interplay of environmental factors and amendment properties affect microbial CUE and the mechanisms involved in

non-additivity of decomposing residue mixtures on soil properties (Kuebbing and Bradford, 2019), which need to be accounted for to be able to create a methodology for optimised benefits from crop residues as soil amendments in arable cropping systems. Therefore, future research on residue mixtures should incorporate not only substrate quality, but also application rate (quantity), diversity (number of residue species) and mixing ratio and how these interact with different arable soil types.

**5. Conclusions**

This experiment tested agronomic benefits obtained from multi-component and chemically heterogeneous residue mixtures compared to the individual residues. Significant positive non-additive effects on available N and SOM were measured after application of a straw-compost mixture, so we can partially accept our first hypothesis predicting greater levels of soil available nutrients and SOM in mixtures compared to individual residues. However, due to variation in the total %C contents

across the experimental field site, we have some reservations about this result. Nevertheless, this study provides some evidence for the potential of crop residue mixtures to provide greater agronomic benefits than single high-C residue amendments of straw or woodchips, at least in terms of preventing N immobilisation during crop growth.

**Data availability**

Data have been uploaded on Mendeley Data, doi:10.17632/jcrvmb8hwy.1

**Supplement link**

[See relevant document in our submission]



**Author contributions**

MS and TS designed the experiment and performed field work. MS carried out laboratory work. MS analysed the data with support from TS and AW. MS prepared the manuscript with critical review by all authors. TS secured funding and
established contact with the farm where the experiment was carried out. TS supervised the project and AW and SM co-supervised the project.

**Competing interests**

The authors declare that they have no conflict of interest.

**Acknowledgements**

This work was funded by a University of Reading Faculty of Science/SAGES Studentship awarded to Marijke Struijk, and research expenses provided by the Waitrose Agronomy Group. We thank G's and their employees for supply of materials and access to the field site, Xin Shu and Adetunji Adekanmbi for help in the field, Omar El-Huni and Alfonso Rodriguez Vila for help with lab work, and Anne S. Dudley, Karen J. Gutteridge, Ilse Kamerling, Fengjuan Xiao and Chris Speed for technical assistance in the lab.

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





**Figures**

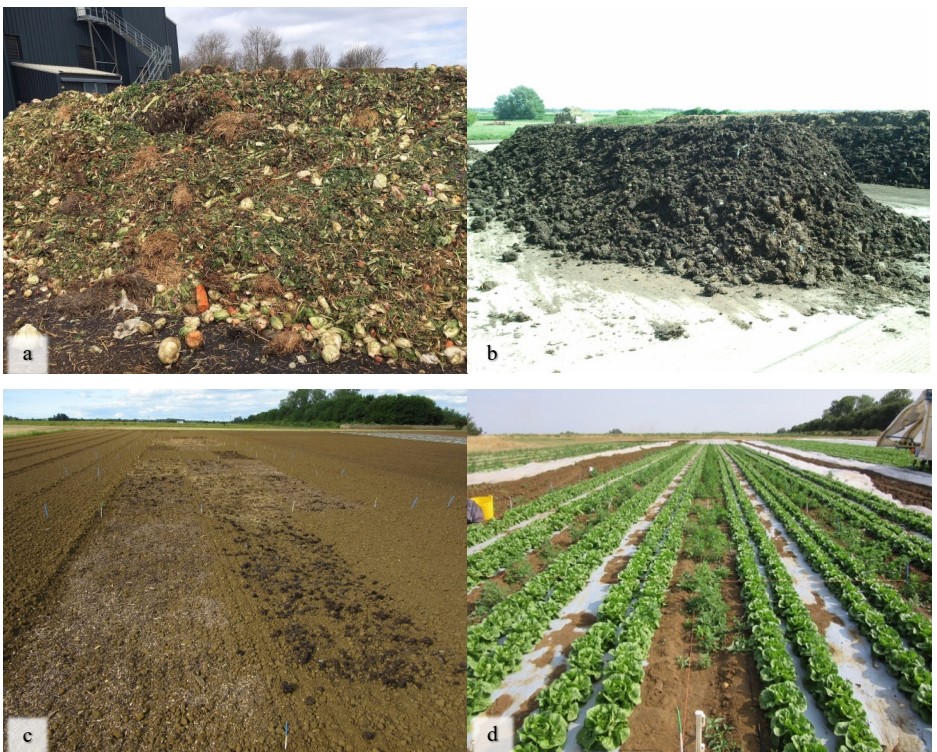

Figure 1: Photographs of the preparation of the mixed compost (a), the final compost product (b), the treatments applied on the experimental plots (c), and the lettuce at time of harvest (d).



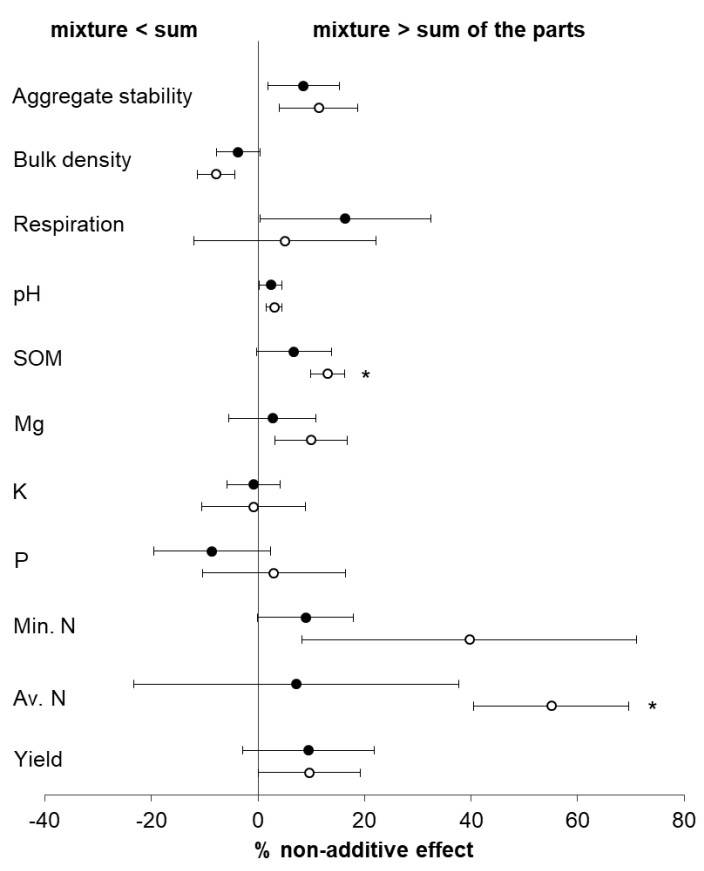

○ straw-compost mixture   ● woodchip-compost mixture

**Figure 2: Non-additive effects of crop-residue mixtures on soil properties. The % non-additive effect is the difference in % effect between the mixture and the sum of the parts. Positive % non-additive effects mean that the effect of the mixture is greater than the sum of the parts, and vice versa. Yield is total lettuce biomass produced per plot, Av. N is available N, Min. N is potentially mineralisable N, soil P, K, and Mg are soil available nutrients, SOM measured as LOI, soil respiration assessed by $CO_2$ burst. Error bars represent SEM (n = 4). Significant difference from zero (where 0 = no significant non-additivity) is indicated by * (one-tailed T-test, $p < 0.05$).**

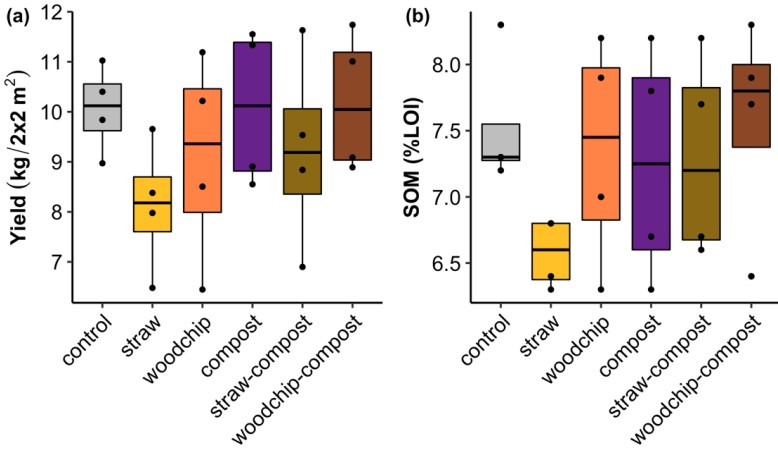

**Figure 3: (a) Gem lettuce yield as total biomass produced per 2 m × 2 m plot sampled. (b) Soil organic matter by percent loss on ignition (% LOI) after each soil amendment treatment. Lower and upper hinges correspond to the 25th and 75th percentiles; black dots represent individual data points, occasionally overlapping (n = 4).**

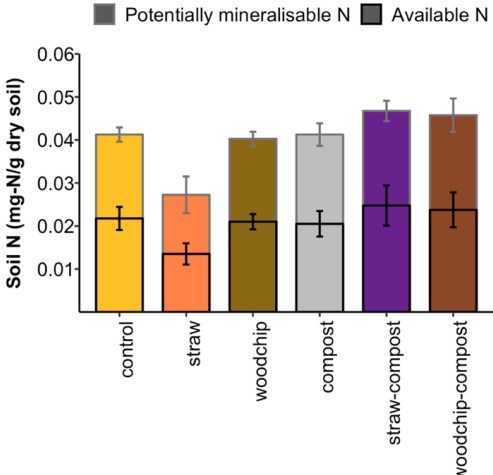

**Figure 4: Soil available and potentially mineralisable N after each soil amendment treatment. Error bars represent SEM of available and potentially mineralisable N separately (n = 4).**





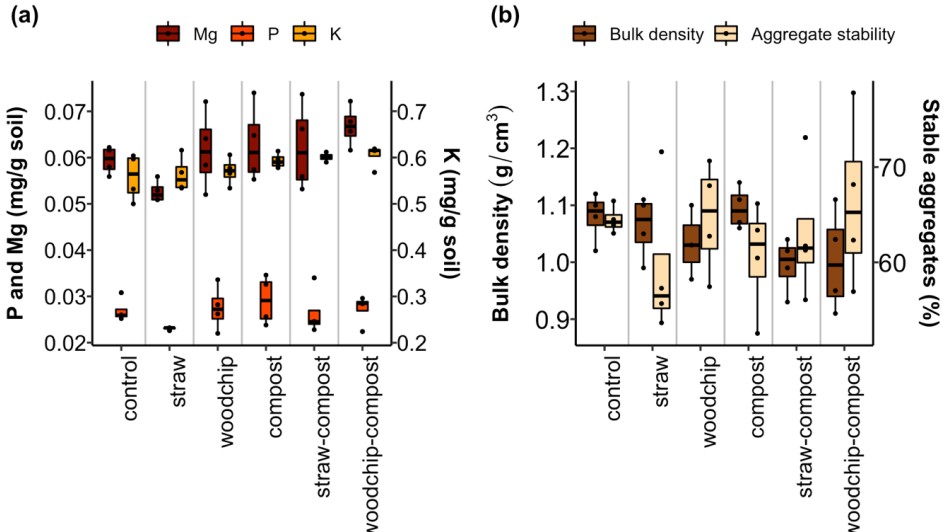

**Figure 5: (a) Soil available nutrients after each soil amendment treatment. (b) Soil physical properties after each treatment. Lower and upper hinges correspond to the 25th and 75th percentiles; black dots represent individual data points, occasionally overlapping (n = 4).**






**Tables**

**Table 1: Treatment structure composed of the factors** *residue* **and** *compost*.

| compost →<br>residue ↓ | **compost** | **no compost** |
|---|---|---|
| **straw** | straw-compost | straw |
| **woodchips** | woodchip-comp | woodchip |
| **none** | compost | control |

**Table 2: Baseline soil data for each treatment (SEM indicated in parentheses, n = 4).**

| | Soil LOI (%) | Soil C:N | Soil pH | Clay content (%) |
|---|---|---|---|---|
| **compost** | 7.94 (0.45) | 10.77 (0.25) | 8.30 (0.03) | 23.3 (0.75) |
| **straw** | 6.84 (0.03) | 10.45 (0.09) | 8.30 (0.05) | 23.8 (1.18) |
| **straw-compost** | 7.78 (0.51) | 10.76 (0.20) | 8.32 (0.04) | 26.0 (1.08) |
| **woodchip** | 8.03 (0.51) | 10.64 (0.39) | 8.27 (0.03) | 24.3 (0.75) |
| **woodchip-compost** | 8.29 (0.47) | 10.95 (0.22) | 8.32 (0.03) | 26.5 (1.26) |
| **control** | 8.14 (0.32) | 10.79 (0.13) | 8.21 (0.02) | 24.5 (1.50) |


**Table 3: Residue characterisation (SEM indicated in parentheses, n = 3).**

| Nutrient | compost | straw | woodchip |
|---|---|---|---|
| C (g/kg) | 322.3 (0.433) | 459.0 (1.012) | 485.3 (1.121) |
| N (g/kg) | 25.3 (0.167) | 11.2 (0.083) | 7.6 (0.105) |
| *C:N* | *12.7 (0.084)* | *40.9 (0.368)* | *63.6 (0.760)* |
| P (g/kg) | 5.5 (0.076) | 1.0 (0.025) | 0.9 (0.024) |
| K (g/kg) | 20.6 (0.31) | 13.1 (0.22) | 5.1 (0.10) |
| Mg (g/kg) | 4.3 (0.014) | 0.7 (0.015) | 1.3 (0.040) |

**Table 4: Amount of C, N and other nutrients applied in each treatment (g/plot).**

| | straw | woodchip | compost | straw-compost | woodchip-compost |
|---|---|---|---|---|---|
| C | 4645 | 5047 | 2707 | 8197 | 7754 |
| N | 114 | 79 | 213 | 347 | 292 |
| *C:N ratio* | *41* | *64* | *13* | *24* | *27* |
| P | 11 | 9 | 46 | 59 | 55 |
| K | 133 | 53 | 173 | 330 | 226 |
| Mg | 7 | 14 | 37 | 45 | 50 |



**Table 5: Statistical outputs of one-tailed T-tests of non-additive effects.** Significance of deviation from additivity (0) is indicated as **p < 0.05** and p < 0.1.

| | straw-compost mixture | | | woodchip-compost mixture | | |
|---|---|---|---|---|---|---|
| | Mean % non-additivity | T | p | Mean % non-additivity | T | p |
| Yield | 9.66 | 1.004 | 0.195 | 9.54 | 0.771 | 0.249 |
| Available N | 55.06 | 3.789 | **0.016** | 7.16 | 0.235 | 0.415 |
| Mineralisable N | 39.67 | 1.265 | 0.147 | 8.93 | 0.990 | 0.198 |
| P | 3.01 | 0.226 | 0.417 | -8.60 | -0.788 | 0.756 |
| K | -0.79 | -0.082 | 0.530 | -0.86 | -0.171 | 0.562 |
| Mg | 9.95 | 1.475 | 0.118 | 2.73 | 0.335 | 0.380 |
| SOM | 13.10 | 4.022 | **0.014** | 6.73 | 0.954 | 0.205 |
| pH | 3.04 | 2.006 | 0.931 | 2.41 | 1.118 | 0.828 |
| Respiration | 5.12 | 0.300 | 0.392 | 16.41 | 1.023 | 0.191 |
| Bulk density | -7.80 | -2.232 | 0.056 | -3.73 | -0.919 | 0.213 |
| Aggregate stability | 11.41 | 1.555 | 0.109 | 8.57 | 1.291 | 0.144 |

**Table 6: Qualitative assessment of lettuce plants as the % of lettuce heads per plot affected by each condition. "Overall" quality impairment is the % of lettuce head per plot affected by one or more conditions. Mean values per treatment (n = 4; SEM in parentheses).**

| Treatment | Chlorosis | | Tip burn | Rot | Overall |
|---|---|---|---|---|---|
| | (All) | (Tips only) | | | |
| control | 49.1 (16.1) | 47.3 (16.9) | 15.5 (4.9) | 1.7 (1.1) | 77.8 (12.3) |
| straw | 31.5 (11.7) | 21.7 (7.49) | 1.9 (1.3) | 0.0 (0.0) | 43.1 (15.6) |
| woodchip | 39.3 (9.3) | 33.4 (8.5) | 12.2 (4.5) | 4.3 (2.0) | 80.4 (11.6) |
| compost | 40.4 (7.7) | 34.3 (6.6) | 14.5 (9.0) | 0.6 (0.6) | 69.4 (10.7) |
| straw-compost | 58.3 (14.9) | 56.1 (15.2) | 16.9 (8.6) | 0.7 (0.7) | 93.0 (7.0) |
| woodchip-compost | 61.7 (14.0) | 54.1 (16.7) | 18.0 (8.7) | 0.0 (0.0) | 82.2 (11.0) |




**Table 7: Selected Pearson correlations (r-values).** Significance indicated as **p < 0.05** and p < 0.10.

| | Yield | Av N | Av+Min N | P | K | Mg | SOM | Resp |
|---|---|---|---|---|---|---|---|---|
| App_C | -0.10 | 0.17 | 0.17 | -0.07 | 0.40 | 0.22 | 0.00 | -0.01 |
| App_N | 0.07 | 0.26 | 0.30 | 0.08 | **0.54** | 0.32 | 0.06 | -0.09 |
| App_P | 0.00 | 0.20 | 0.23 | 0.00 | **0.49** | 0.22 | -0.01 | -0.17 |
| App_K | 0.17 | 0.30 | 0.36 | 0.17 | **0.56** | 0.39 | 0.12 | -0.05 |
| App_Mg | 0.19 | 0.33 | 0.38 | 0.19 | **0.56** | **0.45** | 0.16 | 0.02 |
| Yield | - | 0.29 | **0.45** | **0.75** | 0.19 | **0.78** | **0.74** | 0.36 |
| Av N | 0.29 | - | **0.91** | **0.42** | 0.27 | **0.55** | **0.58** | 0.36 |
| Av + Min N | **0.45** | **0.91** | - | **0.49** | 0.35 | **0.61** | **0.65** | 0.30 |
| P | **0.75** | **0.42** | **0.49** | - | 0.02 | **0.83** | **0.86** | **0.51** |
| K | 0.19 | 0.27 | 0.35 | 0.02 | - | 0.35 | 0.02 | -0.26 |
| Mg | **0.78** | **0.55** | **0.61** | **0.83** | 0.35 | - | **0.80** | **0.47** |
| SOM | **0.74** | **0.58** | **0.65** | **0.86** | 0.02 | **0.80** | - | **0.62** |
| Agg stab | **0.45** | **0.55** | **0.48** | 0.36 | 0.00 | **0.58** | **0.58** | **0.41** |
| Overall qual. | 0.20 | **0.51** | **0.46** | 0.10 | 0.34 | 0.23 | 0.21 | 0.14 |

Abbreviations: App_ = application rate of, Av = available, Min = potentially mineralisable, Agg stab = aggregate stability, Resp = soil respiration, Overall qual. = overall quality impairment.