# Peer review of "Obtaining more benefits from crop residues as soil amendments by application as chemically heterogeneous mixtures"

_SOIL, 2020_

## Referee Comment (RC1) · Anonymous Referee #1 · 14 Apr 2020

General comments

This manuscript reports the results of a full-factorial field experiment, carefully conducted to examine the short-term effects of different mixtures of crop residues on lettuce crop yield and soil respiration, aggregate stability, bulk density, and organic C, N, P, K and Mg contents. The authors found significant non-additive increases in soil available N and soil organic matter contents with the application of a straw-compost mixture (stronger effects than those of straw and compost applied separately) 44 days after the application of the amendments. The topic is important to improve management strategies for crop residues, and the results are well discussed in general. However,

some aspects detailed below would need to be clarified before publication (especially aspects related to the first hypothesis and the methodology).

Specific comments and technical suggestions

L. 13. "(i.e. mixture $\neq$ sum of the parts)" is not needed.

L. 47. "(mixture > sum of the parts)" is also redundant and not needed.

L. 64. What "other soil properties" are the authors referring to?

L. 113-115. The authors hypothesized that faster decomposition of the mixtures would result in higher soil respiration and greater levels of SOM. To fairly test this hypothesis, it would be necessary to differentiate between SOM and amendment organic matter, and this study fails to do so (SOM measurements included amendment organic matter). In other words, it is not possible to tell how much of the OM added with the amendment has been transformed into SOM.

L. 134. Is +-0.8 a standard error? Why only for straw?

L. 134. More information about the amendments would be very useful, especially about the preparation of the compost (stability and maturity).

L. 137-139. "roughly twice". Did the authors consider this error (and those of the application of the individual amendments) when evaluating additive effects?

L. 140. To what depth were the residues incorporated in the soil profile?

L. 140. Was the "fresh" compost (l. 134) mature enough for sowing just the day after its application?

L. 150. Table 2 can be moved to supplementary information.

L. 151. How long was the composting process? Did you perform any maturity or stability test? What was the average size of the straw and woodchip amendments?

L. 182. LOI at 430 $^{\circ}$C instead of 500 $^{\circ}$C as above for residue characterization. Why?

L. 233. The text "some of which were statistically significant (Table 5). Most notably…" is too vague. Only the effects on available N and SOM were significant. Please reword for clarity.

L. 252-254. I agree with the statement that "measurements from residue-mixture treatments cannot be directly compared to individual-residue treatments." Hypothesis 1, however, is stated as "faster decomposition of residue mixtures will result in a higher soil respiration rate in the short term, as well as the release of greater levels of soil available nutrients (N, P, K, Mg) and SOM compared to individual residues." Please reword to clarify.

L. 281 and S3. How were these percentages calculated and on which assumptions? Soil cores used for nutrient analysis were collected to 20-cm depth, whereas bulk density was measured on separate 10-cm-depth soil cores (and the authors still need to specify to what depth the residues were incorporated).

L. 312. Again hypothesis 1. Here decomposition is measured in terms of soil respiration, so how can it be that faster decomposition of an organic input compared to another results in higher soil respiration and greater levels of soil organic matter at the same time?

L. 322. Do you mean that the Solvita burst method is not accurate enough? Please clarify.

L. 322. You mentioned above that soils were sieved to 2 mm, and here that they were sieved to 4 mm. Please clarify.

L. 364-365. Organic amendments could stimulate native soil organic matter mineralization while increasing total soil organic matter content. Why not?

Table 5 could be moved to supplementary information.

---

## Referee Comment (RC2) · Flavia Pinzari (Referee) · 28 Apr 2020

Transferring knowledge on the decomposition of forest litter to the agricultural sector, to improve organic inputs to the soil by burying plant residues, is certainly an exciting and worthwhile approach. This study is very accurate, scientifically sound and is carried out with great care. The manuscript is well written, and the introduction is very exhaustive. A few comments and some suggestions that could help to improve the clarity of some parts are provided.

Introduction The authors well referred to the mechanisms that contribute to the synergistic and not merely additive effects of a mixture of residues. It would be useful to have here a mention of the role of the diversity of decomposers and their niches, and what emerged in studies on forest systems regarding the composition of communities in homogeneous systems compared to mixed systems, not limiting the scenario as purely a matter of microbial carbon use efficiency. The article https://doi.org/10.1111/mec.13739 and other (some quoted therein) give some useful perspectives on dynamics (and complex succession) of bacteria and fungi during decomposition.

Materials and methods The authors limited the analysis of chemical elements to K, P and Mg, besides C and N. The decomposition of organic residues is also linked to other elements that play an essential role in some enzymatic activities fundamental in oxidative processes (manganese, copper and iron, for example). I find this being a bit of a limit of the work that indeed found some significant correlations, reported in the last table and only marginally commented. The introduction of specific residues may have had an impact on the availability of certain microelements capable of explaining the course of some processes.

Results Lines 240-45: "Contrary to our hypothesis, there was a non-additive increase in pH from the mixtures relative to individual amendments (hypothesis 1)". Why should pH increase more in a synergic scenario? How would the authors link pH dynamics with decomposition and the quality of starting materials? This part needs clarifications.

Correlations 3.3- This part would deserve more space. Although it is true that the effect of the synergy of the use of diversified starting materials can be well seen on the quantitative data, I believe that correlations between the amount of nutrients applied and the amount of available K and Mg in the soils are extremely significant. I would be curious to see if the values of R and its sign (+/-) changes when calculated between respiration or SOM and K, Mg, P within each treatment, or between the two main kinds of treatments (mixed or not).

Discussion The different organic residues may have contributed groups of microorganisms capable of directing the decomposition process. The effect of the founder is a process that has a certain weight in the phenomena of ecological succession, especially in the case of microorganisms (see, for example, this work: http://dx.doi.org/10.1016/j.pedobi.2017.01.004). The manuscript under revision is very articulated and definitely not focused on the more strictly microbiological aspects, but a mention of the possible role of the microbial charge of the source material is necessary for the discussion of the results.

---

## Author Comment (AC1) · 11 Jun 2020

Response to Anonymous Referee #1 (RC1)

**General comments**
**This manuscript reports the results of a full-factorial field experiment, carefully conducted to examine the short-term effects of different mixtures of crop residues on lettuce crop yield and soil respiration, aggregate stability, bulk density, and organic C, N, P, K and Mg contents. The authors found significant non-additive increases in soil available N and soil organic matter contents with the application of a straw-compost mixture (stronger effects than those of straw and compost applied separately) 44 days after the application of the amendments. The topic is important to improve management strategies for crop residues, and the results are well discussed in general. However, some aspects detailed below would need to be clarified before publication (especially aspects related to the first hypothesis and the methodology).**

Thank you for the thorough and detailed review of our manuscript. Please see below for our responses to each of your comments by line number.

**Specific comments and technical suggestions**
**L. 13. "(i.e. mixture 6= sum of the parts)" is not needed**
We agree to remove this from the sentence so that it now reads:

> *"Mixing high C:N ratio with low C:N ratio amendments may result in greater carbon use efficiency and non-additive benefits in soil properties (i.e. mixture ≠ sum of the parts)."*

**L. 47. "(mixture > sum of the parts)" is also redundant and not needed.**
We agree to remove this from the sentence so that it now reads:

> *"Synergistic non-additive mixing effects are frequently observed, i.e. decomposition of the mixture is greater than would be predicted from the rate of decomposition of individual litter types (mixture > sum of the parts), especially when the litters are chemically heterogeneous (Pérez Harguindeguy et al., 2008; Wardle et al., 1997)"*

**L. 64. What "other soil properties" are the authors referring to?**
Since the sentence prior to this one only makes reference to C and N we agree with the point that the reviewer has highlighted and agree to revise the following sentence:

> *"These experiments suggest that non-additivity in decomposition rates and changes to other soil properties could go hand-in-hand."*

so that is now reads:

*"These experiments suggest that non-additivity in decomposition rates and changes to soil C and N dynamics could go hand-in-hand"*

**L. 113-115. The authors hypothesized that faster decomposition of the mixtures would result in higher soil respiration and greater levels of SOM. To fairly test this hypothesis, it would be necessary to differentiate between SOM and amendment organic matter, and this study fails to do so (SOM measurements included amendment organic matter).In other words, it is not possible to tell how much of the OM added with the amendment has been transformed into SOM.**
This is true. However, in this study we assume that in each block the SOM levels measured in the control plots are representative of the levels found if no amendments are added.

**L. 134. Is +-0.8 a standard error? Why only for straw?**
Due to the low density of straw, several buckets had to be weighed out to add up to a rate equivalent to 10 t/ha straw. Woodchips and compost are more dense and were much easier to accurately weigh out. The range of actual straw application rates was within the range of 9.2 t ha$^{-1}$ and 10.8 t ha$^{-1}$, hence 10±0.8 t ha$^{-1}$. We will rephrase this for clarity as follows:
> *"… and 10±0.8 t ha$^{-1}$ straw (equivalent to 9.2±0.8 t ha$^{-1}$ dry matter; **± indicates inclusive range of the straw application rate**)."*

**L. 134. More information about the amendments would be very useful, especially about the preparation of the compost (stability and maturity).**
The compost was not made under laboratory conditions specifically for the experiment. It was prepared by the farm and they have many years of experience preparing compost with their residues. Our primary objective was to combine low C/N ratio residues and high C/N ratio residues readily available on-site. Following the standard practice of the farm, the raw materials (as mentioned in line 153) were collected and mixed on 26 March and composted for 52 days prior to use in the experiment on 17 May when the amendment mixtures (straw-compost and woodchip-compost) were prepared. The amendment mixtures were then left for another 26 days prior to field application, allowing some further maturation of the compost. We have no stability data of the compost, but observed that the compost appeared to be well rotted (see the photograph in Fig. 1b in the manuscript).

**L. 137-139. "roughly twice". Did the authors consider this error (and those of the application of the individual amendments) when evaluating additive effects?**
Thank you for pointing this out. We suggest that the word "roughly" can be removed because it results from the range of application rates addressed in the previous comment (line 134).

**L. 140. To what depth were the residues incorporated in the soil profile?**
After application the residues were incorporated with a power harrow, which mixes the residues with the soil in the top ~15 cm. It may be worthwhile to note that the power harrow does not invert the soil, so the residues were not buried but mixed.

**L. 140. Was the "fresh" compost (l. 134) mature enough for sowing just the day after its application?**
As indicated above in line 134, we followed the farm's experience in using compost at their farm and fitted the experiment in their regular procedure of using compost. Our main interest in using the compost was not related to its maturity but the high N content of this residue (see Table 3). The term 'fresh' in the context used in l. 134 refers to the mass (20 t ha$^{-1}$) being the fresh weight, as opposed to the dry weight (7 t ha$^{-1}$). To avoid confusion, we propose to remove the word 'fresh' from this sentence so that it now reads:

> *Application rates of the different amendments were 20 t ha$^{-1}$  compost (equivalent to 7 t ha$^{-1}$ dry matter), 13.3 t ha$^{-1}$ woodchips (equivalent to 8.7 t ha$^{-1}$ dry matter) and 10±0.8 t ha$^{-1}$ straw (equivalent to 9.2±0.8 t ha$^{-1}$ dry matter).*

**L. 150. Table 2 can be moved to supplementary information.**
Since Table 2 contains baseline soil data that are not essential to understand the findings of this work, we agree to move it to the Supplement.

**L. 151. How long was the composting process? Did you perform any maturity or stability test? What was the average size of the straw and woodchip amendments?**
The composting process took 52 days. No maturity or stability tests were performed. The straw was in stalks of about 10-40 cm and the woodchips were about 1.5 x 1.5 x 0.5 cm. The straw was taken from a well-mixed pile.

**L. 182. LOI at 430ºC instead of 500ºC as above for residue characterization. Why?**
Because the initial soil samples were analysed at the University of Reading, according to our standard in-house protocol, while the later samples were analysed at a commercial laboratory (NRM) according to their standard protocol.

**L. 233. The text "some of which were statistically significant (Table 5). Most notably..."is too vague. Only the effects on available N and SOM were significant. Please reword for clarity.**
The sentences that follow the quoted text contain further detail about those effects that were significant. Since this was unclear, we suggest to rephrase the paragraph as follows:

> *"Both compost-residue mixtures resulted in a non-additive increase in lettuce yield, available and potentially mineralisable N, available Mg, SOM, and soil respiration, but not in available K (hypothesis 1) some of which were statistically significant, **as further specified below** (Table 5). Most notably, we observed greater available N and SOM levels in soils to which a mixture of residues was applied, compared to the available N and SOM levels in treatments receiving only individual residue amendments. The straw-compost mixture resulted in a significant (T = 4.022, p = 0.014) non-additive increase in SOM of 13.10%, and while the woodchip-compost mixture did not result in statistically significant non-additivity (T = 0.954, p = 0.205), it did result in a positive non-additive increase in SOM of 6.73%."*

**L. 252-254. I agree with the statement that "measurements from residue-mixture treatments cannot be directly compared to individual-residue treatments." Hypothesis 1, however, is**

**stated as "faster decomposition of residue mixtures will result in a higher soil respiration rate in the short term, as well as the release of greater levels of soil available nutrients (N, P, K, Mg) and SOM compared to individual residues." Please reword to clarify.**
H1 predicts a non-additive increase in soil respiration, but this was not the case. It is different from directly comparing treatments, as the treatments were set up to enable calculation of non-additive effects, not to compare individual treatments.

To clarify we will reword the hypothesis (L. 114) so that it reads:
> *"In particular, we hypothesised faster decomposition of residue mixtures to result in a higher soil respiration rate in the short term, as well as the release of greater levels of soil available nutrients (N, P, K, Mg) and SOM compared to **what would be expected by combining the effects of** individual residues (hypothesis 1)"*

**L. 281 and S3. How were these percentages calculated and on which assumptions? Soil cores used for nutrient analysis were collected to 20-cm depth, whereas bulk density was measured on separate 10-cm-depth soil cores (and the authors still need to specify to what depth the residues were incorporated).**
Thank you for pointing out incomplete calculations. The revised calculations are as follows and will be included in the relevant section of the Supplement:

First the mass of nutrients applied per plot was calculated, using the application rate of each residue (kg residue/plot) and the amount of each nutrient in the residues (mg nutrient/kg residue):

$$Nutrients\ applied\ (mg\ plot^{-1}) = Reisude\ applied\ to\ each\ plot\ (kg\ plot^{-1}) \times nutrient\ content\ of\ reisdues\ (mg\ kg^{-1})$$

Then, using the plot volume to 20 cm depth 0.2 m $\times$ 6 m $\times$ 2 m = 2.4 m$_3$/plot) and the bulk density (g/m$_3$), assuming the bulk density is representative of the top 20 cm and assuming nutrients from the residues applied remained in the top 20 cm of the soil (the sampling depth), we calculated the amount of nutrients added per g of soil via the residues (mg nutrients/g soil) as:

$$Nutrient\ application\ rate\ (mg\ g^{-1}) = \frac{Nutrients\ applied\ (mg\ plot^{-1})}{Area\ of\ the\ plot\ (m^3)}/Bulk\ density\ (g\ m^{-3})$$

Then the difference between the amount of nutrients measured in each plot and the average amount of nutrients measured in the control plots was calculated as:

$$Nutrient\ increase\ (mg\ g^{-1}) = Nutrients\ in\ treatment\ plot\ (mg\ g^{-1}) - nutreints\ in\ control\ plot\ (mg\ g^{-1})$$

Then we determined the increase in soil available nutrients (relative to control) as a proportion of the amount of nutrients added to the soil via residue amendments:

$$Increase\ in\ available\ soil\ nutrients\ (\%) = \frac{Nutrient\ increase\ (mg\ g^{-1})}{Nutrient\ application\ rate\ (mg\ g^{-1})} \times 100\%$$

**L. 312.  Again hypothesis 1. Here decomposition is measured in terms of soil respiration, so how can it be that faster decomposition of an organic input compared to another results in higher soil respiration and greater levels of soil organic matter at the same time?**
Soil samples were sieved to 2 mm before measurements of SOM or respiration were made. Therefore, the > 2 mm pieces of the amendments themselves, although organic, do not directly contribute to the measurement of either SOM, or respiration. We therefore hypothesise that greater respiration and SOM in the residue mixture treatments (compared to single residue treatments) is because the > 2 mm pieces in the residue mixture treatments are decomposed faster and more of the carbon associated with these pieces becomes live microbial biomass, dead microbial necromass, or the products of microbial respiration. We hypothesise that the treatments with only one residue applied decomposed more slowly and so more of the carbon in these samples remains in pieces that are > 2 mm and do not contribute to the SOM or respiration measurements. This hypothesis is related to the understanding that most SOM is derived from microbial processing and the fact that microbes operate at a certain CUE (carbon use efficiency - the fraction of carbon assimilated from organic matter additions to the soil system compared to C losses to the atmosphere via microbial respiration).

**L. 322. Do you mean that the Solvita burst method is not accurate enough? Please clarify.**
We are not criticizing the accuracy of the Solvita burst method. We are instead acknowledging that the method may not be entirely representative of respiration in the field because (1) samples are removed from the field, (2) sample size is very small, and (3) chunks of residues greater than the sieve mesh are sieved out. Soil respiration may be different when measured under field conditions and in un-sieved condition. Moreover, the small sample size may not be representative of the whole soil plot. We can clarify this in line 322 as follows:
> *"Moreover, our soil respiration measurements were taken by the Solvita burst method, on soil samples removed from the field and sieved to **2** mm **removing parts of residues and other organic matter greater than 2 mm,** which may not have been a good representation of the respiration from a soil mixed with crop residues at various stages of decomposition."*

**L. 322. You mentioned above that soils were sieved to 2 mm, and here that they were sieved to 4 mm. Please clarify.**

Thank you for pointing this out. We have double-checked this and the 4 mm is a mistake. It should be 2 mm. Thank you for spotting this.

**L. 364-365. Organic amendments could stimulate native soil organic matter mineralization while increasing total soil organic matter content. Why not?**
Indeed, we agree that organic amendments could stimulate the mineralisation native SOM (priming). However, in line 364-365 we are comparing the SOM level in the straw-compost treatment to the control treatment, rather than assessing the non-additive effect which involves the individual amendment treatments.
We will clarify this by revising as:
> *"Even in the straw-compost treatment, the SOM level was very close to that of the control treatment, suggesting **net** mineralisation of native SOM **as a result of the residue amendment** was negligible"*

**Table 5 could be moved to supplementary information.**
We would be happy to move Table 5 to the Supplement.

We hope we have responded to these comments in a satisfactory manner.

---

## Author Comment (AC2) · 11 Jun 2020

Response to Flavia Pinzari (RC2)

**Transferring knowledge on the decomposition of forest litter to the agricultural sector, to improve organic inputs to the soil by burying plant residues, is certainly an exciting and worthwhile approach. This study is very accurate, scientifically sound and is carried out with great care. The manuscript is well written, and the introduction is very exhaustive. A few comments and some suggestions that could help to improve the clarity of some parts are provided.**
Thank you for the thorough and detailed review of our manuscript. Please see below for our responses to each of your comments.

**Introduction: The authors well referred to the mechanisms that contribute to the synergistic and not merely additive effects of a mixture of residues. It would be useful to have here a mention of the role of the diversity of decomposers and their niches, and what emerged in studies on forest systems regarding the composition of communities in homogeneous systems compared to mixed systems, not limiting the scenario as purely a matter of microbial carbon use efficiency. The article https://doi.org/10.1111/mec.13739 and other (some quoted therein) give some useful perspectives on dynamics (and complex succession) of bacteria and fungi during de-composition.**
We will incorporate the suggestion to include mention of microbial succession during decomposition by adding the following sentence to the paragraph starting at line 72:

"… Other authors have also suggested the possibility of manipulating the functionality of the soil microbial community with soil amendments, such as Li *et al*. (2019) who report that eutrophic microbes are stimulated by organic carbon amendments and oligotrophic microbes are stimulated by chemical fertilisers. Studies have also demonstrated that changes in tree litter diversity affect both fungal and bacterial diversity (Otsing *et al*., 2018; Santonja *et al*., 2018). **Research on decomposition in forest systems indicates a succession in the community composition of microbial decomposers as the decomposition of residues progresses (Bastian *et al*., 2009; Purahong *et al*., 2016), and this succession is different in decomposition of litters of different qualities (Aneja et al., 2006)."**

Additional references to include:
Aneja (2006): doi.org/10.1007%2Fs00248-006-9006-3
Bastian (2009): doi.org/10.1016/j.soilbio.2008.10.024

Puraong et al. (2016): doi.org/10.1111/mec.13739

**Materials and methods: The authors limited the analysis of chemical elements to K, P and Mg, besides C and N. The decomposition of organic residues is also linked to other elements that play an essential role in some enzymatic activities fundamental in oxidative processes (manganese, copper and iron, for example). I find this being a bit of a limit of the work that indeed found some significant correlations, reported in the last table and only marginally commented. The introduction of specific residues may have had an impact on the availability of certain microelements capable of explaining the course of some processes.**
L. 64 commented on by the other reviewer has been revised as follows:

> *"These experiments suggest that non-additivity in decomposition rates and changes to soil C and N dynamics could go hand-in-hand"*

This reviewer comment led us to concede that there is only evidence that non-additivity is related to C and N and not other soil properties. While microelements are important mediators of decomposition in some natural ecosystems (e.g. forests), these agricultural soils undergo frequent analysis by the farm and corrective measured introduced if micronutrients are below optimal levels. Therefore, we have not included them in our discussion.

We have, however, analysed the residues for some of these elements, so we suggest including these in Table 3 as follows:

**Table 3: Residue characterisation (SEM indicated in parentheses, n = 3).**

| Nutrient | compost | straw | woodchip |
|---|---|---|---|
| C (g/kg) | 322.3 (0.433) | 459.0 (1.012) | 485.3 (1.121) |
| N (g/kg) | 25.3 (0.167) | 11.2 (0.083) | 7.6 (0.105) |
| *C:N* | *12.7 (0.084)* | *40.9 (0.368)* | *63.6 (0.760)* |
| P (g/kg) | 5.5 (0.076) | 1.0 (0.025) | 0.9 (0.024) |
| K (g/kg) | 20.6 (0.31) | 13.1 (0.22) | 5.1 (0.10) |
| Mg (g/kg) | 4.3 (0.014) | 0.7 (0.015) | 1.3 (0.040) |
| Mn (mg/kg) | 258 (1.68) | 41 (1.15) | 41 (1.67) |
| Fe (g/kg) | 15.0 (0.051) | 0.5 (0.015) | 1.0 (0.060) |

The recovery of a certified reference material analysed alongside our samples for copper is only 78%. This represents a QC failure and so we are not confident about including the Cu data. The recovery of Mn (92%) and Fe (114%) will be included in the Methods section.

**Results Lines 240-45: "Contrary to our hypothesis, there was a non-additive increase in pH from the mixtures relative to individual amendments (hypothesis 1)". Why should pH increase more in a synergic scenario? How would the authors link pH dynamics with decomposition and the quality of starting materials? This part needs clarifications.**
We agree that clarification is needed because the expected effect of the treatments on soil pH has not been explicitly hypothesised. We will therefore revise the manuscript accordingly.

We would expect pH to increase when organic residues are added to the soil since decomposition of residues leads to the ammonification of residue N (Xu et al., 2006), and since faster decomposition is expected in the mixture treatment, compared to the parts, we would expect a greater increase in pH. Indeed, this should be stated explicitly in hypothesis 1, so we suggest to revise L. 114 to read:

> *"In particular, we hypothesised faster decomposition of residue mixtures to result in a higher soil respiration rate in the short term, as well as the release of greater levels of soil available nutrients (N, P, K, Mg) and SOM, compared to individual residues, which leads to greater ammonification of resudie N (Xu et al., 2006), **and, in turn, leads to a greater increase in pH** (hypothesis 1)."*

This means that the finding that the pH increased is in fact in agreement with hypothesis 1. So we further revise line 240-45 as follows:

> ***"In agreement with*** *our hypothesis, there was a non-additive increase in pH from the mixtures relative to individual amendments (hypothesis 1), although this was not significant (Table 5) and per-treatment results (discussed in next section) show that the pH decreased in all treatments relative to the control (F = 2.238; p = 0.095; one-way ANOVA; Supplement S2)."*

Additional references to include:
Xu et al. (2006): https://doi.org/10.1016/j.soilbio.2005.06.022

**Correlations 3.3 - This part would deserve more space. Although it is true that the effect of the synergy of the use of diversified starting materials can be well seen on the quantitative data, I believe that correlations between the amount of nutrients applied and the amount of available K and Mg in the soils are extremely significant. I would be curious to see if the values of R and its sign (+/-) changes when calculated between respiration or SOM and K, Mg, P within each treatment, or between the two main kinds of treatments (mixed or not).**
We do not wish to undertake correlations of measurements made on replicate plots within an individual treatment since the value of n would be 4 (or 8 if separate correlations are made on replicates in the mixture and non-mixture treatments). It is not clear what hypothesis we would be testing when conducting these correlations.
Significant correlations between the amount of nutrients applied and the amount of available nutrients, and correlations of Yield with available nutrients and SOM are highlighted in section 3.3. However, as requested, we expand on our description of the data presented in Table 7, to also include the significant correlation between SOM and available nutrients:

> "A number of noteworthy correlations may help explain the data and are summarised in Table 7. There were some significant correlations between the amount of nutrients applied and the amount of available K and Mg in the soils at the end of the experiment, which indicates a positive effect of the residue amendments. The amount of C applied via the residue amendments was not correlated with the amount of SOM. Yield was positively correlated with the sum of available and potentially mineralisable N, available P and Mg, SOM and aggregate stability. **SOM was also positively correlated with available N, P and Mg, and with soil respiration.**"

**Discussion: The different organic residues may have contributed groups of microorganisms capable of directing the decomposition process. The effect of the founder is a process that has a certain weight in the phenomena of ecological succession, especially in the case of microorganisms (see, for example, this work: http://dx.doi.org/10.1016/j.pedobi.2017.01.004). The manuscript under revision is very articulated and definitely not focused on the more strictly microbiological aspects, but a mention of the possible role of the microbial charge of the source material is necessary for the discussion of the results.**

Although this does not contribute to the interpretation of our results, it could certainly be mentioned that the residues introduce microbes that may not already be present in the soil. We therefore agree to revise L. 324 as follows:

> *"As pointed out by Lecerf et al. (2011), niche complementarity effects, in which different groups of decomposing organisms **(already present in the soil, or newly introduced via the residues)** develop a synergistic association in residue breakdown, tend to advance with time, leading to a generally higher number of long-term litter-mixing studies finding non-additive effects."*

We hope we have responded to these comments in a satisfactory manner.

---

## Author Response (AR1)

**Point-by-point changes made to "Obtaining more benefits from crop residues as soil amendments by application as chemically heterogeneous mixtures" by Marijke Struijk et al.**

L. 13 (RC1 ) We removed "(i.e. mixture 6= sum of the parts)" so that it now reads: "Mixing high C:N ratio with low C:N ratio amendments may result in greater carbon use efficiency and non-additive benefits in soil properties (i.e. mixture ≠ sum of the parts)."

L. 47 (RC1) We removed "(mixture > sum of the parts)" so that it now reads: "Synergistic non-additive mixing effects are frequently observed, i.e. decomposition of the mixture is greater than would be predicted from the rate of decomposition of individual litter types <del>(mixture > sum of the parts)</del>, especially when the litters are chemically heterogeneous (Pérez Harguindeguy et al., 2008; Wardle et al., 1997)"

**L. 64. (RC1)** We specified the "other soil properties" so that is now reads: *"These experiments suggest that non-additivity in decomposition rates and changes to soil C and N dynamics could go hand-in-hand"*

**L. 92 (RC2)** We incorporate the suggestion to include mention of microbial succession during decomposition by adding the following sentence to the paragraph starting at line 88:

"... Other authors have also suggested the possibility of manipulating the functionality of the soil microbial community with soil amendments, such as Li et al. (2019) who report that eutrophic microbes are stimulated by organic carbon amendments and oligotrophic microbes are stimulated by chemical fertilisers. Studies have also demonstrated that changes in tree litter diversity affect both fungal and bacterial diversity (Otsing et al., 2018; Santonja et al., 2018). **Research on decomposition in forest systems indicates a succession in the community composition of microbial decomposers as the decomposition of residues progresses (Bastian et al., 2009; Purahong et al., 2016), and this succession is different in decomposition of litters of different qualities (Aneja et al., 2006).**"

Additional references were included:

Aneja (2006): doi.org/10.1007%2Fs00248-006-9006-3 Bastian (2009): doi.org/10.1016/j.soilbio.2008.10.024 Puraong et al. (2016): doi.org/10.1111/mec.13739

**L. 118 (RC1 and RC2)** We revised the wording of hypothesis 1 in response to comments from both reviewers. In RC1 there seemed to be some confusion over non-additive effects vs. differences between individual treatments. To clarify we reworded the hypothesis to read:

"In particular, we hypothesised faster decomposition of residue mixtures to result in a higher soil respiration rate in the short term, as well as the release of greater levels of soil available nutrients (N, P, K, Mg) and SOM compared to **what would be expected by combining the effects of** individual residues (hypothesis 1)"

Additionally, details on pH were requested in RC2, so we further revised the hypothesis to read: *"In particular, we hypothesised faster decomposition of residue mixtures to result in a higher soil respiration rate in the short term, as well as the release of greater levels of soil available nutrients (N, P, K, Mg) and SOM, compared to individual residues,* **which**  *leads to greater ammonification of resudie N (Xu et al., 2006), and, in turn, leads to a greater increase in pH* (hypothesis 1)."

The additional reference Xu et al. (2006) has been included in the reference list. This meant we needed to adjust the wording of the pH result in **line 240-45** as follows:

**"In agreement with** our hypothesis, there was a non-additive increase in pH from the mixtures relative to individual amendments (hypothesis 1), although this was not significant (Table 5) and per-treatment results (discussed in next section) show that the pH decreased in all treatments relative to the control (F = 2.238; p = 0.095; one-way ANOVA; Supplement S2)."

**L. 140 (RC1)** To avoid confusion over maturity of compost, we removed the word 'fresh' from this sentence so that it now reads:

Application rates of the different amendments were 20 t  $ha^{-1}$  fresh compost (equivalent to 7 t  $ha^{-1}$  dry matter), 13.3 t  $ha^{-1}$  woodchips (equivalent to 8.7 t  $ha^{-1}$  dry matter) and 10±0.8 t  $ha^{-1}$  straw (equivalent to 9.2±0.8 t  $ha^{-1}$  dry matter).

L. 141 (RC1) The use of ±0.8 has been clarified:

"... and  $10\pm0.8$  t ha-1 straw (equivalent to  $9.2\pm0.8$  t ha-1 dry matter;  $\pm$  indicates inclusive range of the straw application rate)."

**L. 144 (RC1)** We removed the word "roughly" because it results from the range of application rates addressed previously and is therefore redundant.

**L. 157 (RC1)** Table 2 has been moved to supplementary information. All table numbers have been adjusted accordingly both in the tables and in the text.

**L. 170 (RC2)** Additional analytes Mn and Fe were added to the residue characterisation table (originally Table 3, but now renumbered to Table 2) in response to RC2. The recovery rates of these analytes were added to the Methods section.

**L.259. See L.118 above**

L. 248 (RC1) For clarity we rephrased the paragraph as follows:

"Both compost-residue mixtures resulted in a non-additive increase in lettuce yield, available and potentially mineralisable N, available Mg, SOM, and soil respiration, but not in available K (hypothesis 1) some of which were statistically significant, **as further specified below** (Table 5). Most notably, we observed greater available N and SOM levels in soils to which a mixture of residues was applied, compared to the available N and SOM levels in treatments receiving only individual residue amendments. The strawcompost mixture resulted in a significant (T = 4.022, p = 0.014) non-additive increase in SOM of 13.10%, and while the woodchip-compost mixture did not result in statistically significant non-additivity (T = 0.954, p = 0.205), it did result in a positive non-additive increase in crease in SOM of 6.73%."

**L. 317 (RC2)** As requested, we expanded on our description of the data presented in Table 7, to also include the significant correlation between SOM and available nutrients:

"A number of noteworthy correlations may help explain the data and are summarised in Table 7. There were some significant correlations between the amount of nutrients applied and the amount of available K and Mg in the soils at the end of the experiment, which indicates a positive effect of the residue amendments. The amount of C applied via the residue amendments was not correlated with the amount of SOM. Yield was positively correlated with the sum of available and potentially mineralisable N, available P and Mg, SOM and aggregate stability. **SOM was also positively correlated with available N, P and Mg, and with soil respiration.**"

**L. 350 (RC1)** We clarified why Solvita burst method may not be representative of in-situ respiration by adding the following in line 350, and we corrected 4 mm as the sieve size which should have been 2 mm:

"Moreover, our soil respiration measurements were taken by the Solvita burst method, on soil samples removed from the field and sieved to **2** mm **removing parts of residues and other organic matter greater than 2 mm**, which may not have been a good representation of the respiration from a soil mixed with crop residues at various stages of decomposition."

**L. 355 (RC2)** In response to RC2 we made mention of the fact that the residues introduce microbes that may not already be present in the soil by revising in line 355 as follows:

"As pointed out by Lecerf et al. (2011), niche complementarity effects, in which different groups of decomposing organisms **(already present in the soil, or newly introduced via the residues)** develop a synergistic association in residue breakdown, tend to advance with time, leading to a generally higher number of long-term litter-mixing studies finding non-additive effects."

L. 397 (RC1) In response to RC1 we revised as follows:

"Even in the straw-compost treatment, the SOM level was very close to that of the control treatment, suggesting **net** mineralisation of native SOM **as a result of the residue amendment** was negligible"

L. 482. An acknowledgement was added by one of the authors.

**Supplement**

Table S1 was moved from the main text to the supplement in response to RC1, and the rest of the supplement renumbered accordingly.

Table S2 was originally Table 5 and was moved to the supplement in response to RC1. Calculations in S4 were revised and completed.

[revised manuscript text omitted]